# Can Family Involvement Improve Business Performance? Based on the Dual Moderating Effects of Overseas Experience and Charitable Donations

**Shuqin Song** [1,2], **Mengyun Wu** [2], **Yuqing Zhu** [2,*] and **Yihan Lv** [2]

1  Personnel Department, Jiangsu University, Zhenjiang 212013, China
2  School of Finance and Economics, Jiangsu University, Zhenjiang 212013, China
*  Correspondence: 2212019050@stmail.ujs.edu.cn

**Abstract:** In the post-pandemic era, it will become normal for family enterprises to seek innovative development. They have become more committed to building socially responsible companies and are more willing to actively promote corporate values in order to create long-term benefits. Therefore, this paper uses listed family companies entering the intergenerational succession period from 2018 to 2020 as the research object and empirically tests the influence of family involvement on firm performance, as well as the moderating effects of family members' overseas experiences and family firm's charitable donation behaviors. The results show that the ownership and management involvement of family members can significantly improve the performance of family enterprises. The overseas experience of family members has a negative moderating effect on the relationship between the two, while the charitable giving behavior of family enterprises has a positive moderating effect on the relationship between the two. The above research provides countermeasures and reference for family enterprises to realize the family business and the lasting development in the process of intergenerational inheritance, upgrading, and transformation.

**Keywords:** family business; family involvement; overseas experience; charitable giving; sense of social responsibility





## 1. Introduction

Chinese family enterprises were born in a special historical background and shoulder a more ambitious mission of The Times. As the most active component of the private economy, family enterprises have experience since childhood from weak to strong process. Family entrepreneurs rely on trust support, blood relationships, and common value orientation to quickly establish enterprises, providing full play to the entrepreneurial spirit, and casting the legendary story of Chinese family entrepreneurship. As one of the world's top 20 pharmaceutical companies, Boehringer Ingelheim, a family-owned pharmaceutical company, relies on its outstanding performance in the fields of long-term strategic positioning, organizational culture, and talent development to achieve the goal of "becoming rich for three generations". Employees' long-term practices of family values include sustainable development and the driving effect of a diverse and inclusive organizational culture on innovation. The vision of long-term sustainable development is an important force to enhance organizational resilience in the face of uncertain environments. However, in recent years, with the exposure of a series of events, such as the failure of WINDMILL Group Limited and the conflict between the father and son of SHINEWAY, it has been revealed that the current situation of family enterprises is not a strong sense of social responsibility, which has seriously affected the outside world's view of family enterprises and hindered the progress of family enterprises. Therefore, how to realize the long-term stable development of Chinese family businesses has attracted the attention of domestic and foreign scholars.

Family involvement, as a typical symbol that distinguishes family enterprises from ordinary enterprises, is completely different in terms of leadership style and management style and different degrees of family involvement will have different impacts on enterprises [1]. Among them, the influence of family involvement on business performance has also become a hot topic among scholars. At the same time, family members' overseas studies or work backgrounds and the charitable behavior of family businesses pursuing non-economic benefits have also drawn wide attention. On the one hand, most family enterprises are about to enter the period of power transition and the school or work experience of family members will affect the willingness of future successors of the enterprise to take over and then affect the development of the economic performance of the enterprise [2,3]. On the other hand, since the 18th National Congress of the CPC, the Chinese government has emphasized the importance of enterprises practicing social responsibility and that charitable donation is the highest form of social responsibility. Through the implementation of altruistic measures such as charitable donation activities, family enterprises can not only reap economic benefits, but also shape the image of caring enterprises and gain social reputation, which can also affect the value of enterprises subtly [4,5]. This paper attempts to answer the following questions: How does family involvement affect firm performance? Do the overseas experiences of family members and the charitable giving of family firms influence these two factors? As mentioned above, this study will put forward relevant hypotheses based on the review of the previous literature and empirically analyze the data of China's Shanghai–Shenzhen A-share family listed companies to explore the relationship between them, to provide relevant countermeasures for reference for the future sustainable development of family enterprises.

The contributions of this study are as follows: first, the definition of family involvement on the members of the family ownership and management rights involved in double dimensions with the method of measurement, by discussing family ownership and the management level of enterprise performance, the influence of rich research dimensions as well as the further power efficient allocation of family business management power; second, through empirical research on the mechanism of the second generation's involvement in ownership and management rights affecting the performance of family businesses, it is found that the degree of contribution of the second generation's involvement in family businesses and charitable donations can positively promote the development of family businesses and that effective charitable donations can better help family businesses establish their image, win reputation, and gain social recognition; third, from the perspective of background characteristics of family members and behavioral characteristics of family business, the effects of family involvement and firm performance are further discussed and the research on related mechanisms of family business is expanded; finally, using the relationship between the degree of involvement of the second-generation power elements and the target performance of family businesses, and including overseas experience and charitable donations as moderating factors, to enrich research in the related fields and provide a reference for family business practice.

## 2. Literature and Research Hypotheses

### 2.1. Family Involvement and Enterprise Performance

While family involvement injects special vitality into the enterprise, it is also accompanied by conflicts of interest. With the deepening of the involvement of family management power, special agency problems mixed with internal goal conflicts of family enterprises have gradually emerged. The principal-agent theory holds that due to information asymmetry between the owner and the operator, there are conflicts between the two parties in terms of goals and motivations [6]. At the same time, the high concentration of family business ownership and management rights has weakened the traditional agency problem [7]. Based on the principal-agent theory, self-agency in the context of family business still has conflicts and the decision preferences derived from multiple objectives will lead to differences at the level of corporate strategy [8]. In the system of family and business

integration, the methods of family involvement are also diverse. On the one hand, it can control the enterprise through explicit behaviors such as holding the ownership of the family enterprise, intervening in the daily operation and management of the family enterprise, and designating family members as heirs of the enterprise. On the other hand, it can also influence enterprise management through family enterprise culture, political association of family founders, corporate reputation, and other hidden resources [9].

Firstly, based on principal-agent theory and resource-based theory, family involvement can effectively reduce the first type of agency costs, help business leaders obtain family special resources and social and emotional wealth, and help improve business performance. Based on the perspective of corporate governance, when family members serve on the board of directors, the financial performance of enterprises can be significantly improved [10]. In the process of intergenerational transmission, the ownership and management rights of enterprises are concentrated on family members, which can better coordinate the principal-agent problem and improve the efficiency of corporate governance [11]. The high concentration of ownership of family members would cause them to increase their right of discourse in business decision-making, which could not only realize the continuous growth of family business interests, but also improve personal value [12]. The higher the degree of family involvement and the more concentrated the ownership structure, the more innovative activities will be carried out by the enterprise to improve the operating efficiency and finally improve the enterprise performance [13].

Secondly, different from ownership involvement, family management involvement can more directly and effectively reflect the influence and executive ability of the family. However, the management involvement of family members affects the behavior in the daily operation process of family enterprises to a certain extent, which is generally reflected by family members acting as the chairman, general manager, and other top management personnel [14]. Family members can better realize information exchange by mastering management rights, understand the working ability and management ability of different members, and reduce the problem of information asymmetry [15]. They closely combine their personal interests with the interests of the enterprise, adopt the healthy and sustainable development of the family business as their own responsibility, and have a stronger sense of responsibility and diligence. On the contrary, when the CEO has a small management power, it is easy to have financing difficulties, stock price volatility, and financing difficulties [16]. According to the reputation hypothesis theory, the second generation of the family with management rights will be more willing to increase the environmental cost input, establish a green family enterprise image, and improve the environmental performance. Accordingly, the first set of hypotheses are proposed:

**Hypothesis 1a (H1a).** *The involvement of family members in ownership contributes to the improvement of corporate performance.*

**Hypothesis 1b (H1b).** *The involvement of family members in management power contributes to the improvement of corporate performance.*

*2.2. The Moderating Effect of Family Members' Overseas Experience on Family Involvement and Firm Performance*

With the development of economic globalization, more and more family business owners are willing to send their children abroad for further education. About 40% of successors in intergenerational family businesses have overseas study backgrounds [17]. Although scientific and systematic enterprise operation knowledge and advanced management experience can help improve the level of corporate governance, there are certain risks [18]. Firstly, according to a study by CAI's team at Xiamen University, members with overseas experience will be more inclined to the enterprise strategy of diversification and cross-industry mergers and acquisitions after entering the management level, but this series of behaviors cannot obtain high returns in the short term, which is not conducive to improv-

ing enterprise performance [19]. Secondly, family members with overseas backgrounds have low emotional commitment and organizational recognition to the enterprise, weak family cultural awareness, and weak senses of social responsibility, which may lead to reckless investment in order to maximize profits, resulting in the risk of inefficient investment. Finally, under the influence of "differential order pattern," Chinese families prefer "rule by man" rather than illegal rule and rely more on trust between people [20]. The implementation of Western governance methods by family members with overseas background in enterprises can easily cause resistance among insiders and subordinates, which is not conducive to improving enterprise performance [21]. Therefore, this paper believes that the overseas experience of second-generation members will reduce the performance of enterprises. Based on the above analysis, the second set of hypotheses are proposed:

**Hypothesis 2a (H2a).** *Family members' overseas experience has a significant negative regulating effect on family ownership involvement and firm performance.*

**Hypothesis 2b (H2b).** *Family members' overseas experience has a significant negative regulating effect on family management involvement and firm performance.*

### 2.3. The Moderating Effect of Family Charitable Giving Behavior on Family Involvement and Firm Performance

Charitable donation is an important means for family enterprises to fulfill their social responsibilities. Long-term family charitable investment can not only demonstrate the sense of responsibility of the wealthy family, but also gain social recognition and respect for the family [22]. When family members join the board of directors, they will increase the input of charitable donations [23]. A team from the Zhejiang University investigated a large sample of family businesses in the Zhejiang Province and found that about 80% of CEOs who had contributed charitable donations had experienced poverty in their youth and were more willing to engage in charity and public welfare and help others [5]. Therefore, compared with other enterprises, the experience of poverty among family entrepreneurs can promote their children's social responsibility. The establishment of a charitable foundation or charity organization by a family can help the next generation or even the next generation understand the family culture and values that the founder wants to convey, so as to better help future generations cultivate social responsibility, improve the level of family governance, and realize the sustainable inheritance of a family spirit [24]. A legacy of family philanthropy can increase family solidarity better than a legacy of business.

On the other hand, charitable donation behavior can reduce the adverse impact of negative events on the enterprise and maintain the corporate image and reputation when the enterprise has a business crisis [25]. Charitable donations can help enterprises accumulate moral capital and play the role of reputation insurance when enterprises suffer adverse effects [26,27]. This paper argues that in the enterprises with family charitable donation experience, family enterprises can improve their social responsibility and corporate performance more. Based on the above analysis, the third set of hypotheses are proposed:

**Hypothesis 3a (H3a).** *Family charitable giving behavior has a significant positive moderating effect on the relationship between family ownership involvement and firm performance.*

**Hypothesis 3b (H3b).** *Family charitable giving behavior has a significant positive moderating effect on the involvement of family management rights and firm performance.*

## 3. Research Design

### 3.1. Sample Selection and Data Sources

In this paper, family companies listed on China's Shanghai and Shenzhen A-shares from 2018 to 2020 were selected as the research objects and a total of 1371 family listed

companies were initially selected by using "China Private Listed Companies Database" in CSMAR database. After that, the samples were screened as follows: (1) at least one or more family members held posts in the board of directors and supervisors; (2) excluding listed companies marked as ST and ST*; (3) excluding listed financial and insurance companies; (4) excluding listed companies with serious missing main data; and, finally, 3690 valid values of 1230 valid family business samples are obtained, which are 3-year balanced panel data. The data of sample companies used in this paper are mainly from the CSMAR database and the Wind database and network tools such as Flush Finance and the Baidu search engine are used to improve the information of second-generation family members. The basic calculation of the data in the paper was performed using Excel 2016 and the final regression processing was performed using Stata15.0.

*3.2. Variable Definition*

(1)    Explained variable: Enterprise performance

In this paper, the Tobin's Q value, which can represent the market value of the enterprise, is used to measure the enterprise performance.

(2)    Explanatory variables: FO and MO

Family involvement, as one of the distinctive features of family businesses, refers to the continuous investment of human capital, cultural capital, social capital, and financial capital in the family business by the family and different control over the ownership, control, operation, income, culture, etc. of the family business. These influence and produce a series of behaviors or concepts involving family members or family interest communities in the family business, which ultimately affects the growth and development. There are two ways of family involvement, formal involvement and informal involvement. Formal involvement refers to ownership involvement, control right involvement, and operation (management) involvement, etc. Informal involvement includes family willingness involvement, family spirit involvement, etc. Scholars at home and abroad generally use formal family involvement as indicators. Chen Ling and others believe that family involvement mainly refers to the involvement of ownership and management rights, which generally increase the complexity and uncertainty of family business development and may cause lead to fundamental changes in corporate structure, culture, and systems. On the other hand, Jiang [28] judged that when a family member or several family members' children, sons-in-law, and nephews/nieces appeared in the enterprise, family involvement behavior occurred, marked as 1; otherwise, it is marked as 0. Combining two common research methods, the researchers believe that the situation of family involvement includes multiple combinations such as husband and wife joint management, parent–child inheritance, etc., combined with the situation of family business ownership and control, the shares of all family members in the company The ratio of family ownership obtained from the sum of the sum/total shares of the enterprise and the ratio of family management rights obtained from the number of family members in management/total number of managers are more in line with the actual situation.

In this paper, the actual equity of listed companies owned by family members is selected to measure family involvement ownership (FO). Family involvement in management is measured by the proportion of family members working in the management level of the enterprise to the total number of the management level (MO).

$$FO = \frac{\text{Equity in a listed family company owned by family members}}{\text{Total equity of the family listed company}}$$

$$MO = \frac{\text{Number of family members in management positions}}{\text{Total number of management of family listed companies}}$$

(3)    Moderating variables

Whether family members have overseas experience is the first moderating variable in this paper. By consulting the annual reports of listed companies, Flush Finance and other network tools, we manually collect whether the family members serving in the board of directors, the board of supervisors, and the senior management team have overseas study or work backgrounds. If so, the value is 1, otherwise it is 0. Charity is the second moderating variable in this paper. By reviewing the social responsibility reports of listed companies and news media reports, it is manually collected whether the family company has ever donated to charity and public welfare. If so, it is assigned a value of 1, otherwise it is 0.

(4)　Control variables

The variables selected in this paper mainly include: enterprise scale, asset–liability ratio, enterprise age, and enterprise growth. In the aspect of corporate governance structure, the selected variables are measured by the closeness between the government and enterprise and the degree of equity balance. In terms of external investment, the shareholding ratio of enterprise institutional investors is selected for measurement. The relevant definitions of all variables in this paper are shown in Table 1:

**Table 1.** Description and definition of variables.

| Variable Type | Variable Name | Variable Symbol | Variable Definition |
|---|---|---|---|
| Explained variable | Enterprise performance | Tobin-q | Tobin-q = Market value/Assets |
| Explanatory variables | Family involvement ownership | FO | Effective ownership of a public company owned by family members |
| | Family involvement in management | MO | The proportion of family members in corporate management in the total number of corporate managements |
| Moderating variables | Family member overseas experience | Oversea | According to the manually collected information, the value is assigned to 1 and no value is 0 |
| | Corporate charitable donation behaviors | Charity | According to the manually collected information, the value is assigned to 1 and no value is 0 |
| Control variables | Enterprise scale | Size | Use the log of total assets |
| | Asset–liability ratio | Lev | Total liabilities/Total assets |
| | Enterprise age | Age | Use the logarithm of the time of establishment |
| | Enterprise growth | Growth | Growth rate of operating income |
| | The closeness between government and enterprise | Index | The total score of government intervention enterprises in each region in China's Marketization Index Report by Province (2018) is used to measure the close relationship between local government and enterprise |
| | The degree of equity balance | Ebd | The sum of the shareholding ratio of the second to the tenth shareholder/the shareholding ratio of the largest shareholder |
| | Shareholding ratio of institutional investors | Inde | Institutional holdings/Total shares |

### 3.3. Model Construction

On the basis of the above theoretical analysis, this paper carries out the test model design in two steps.

Step one: examining the influence of family involvement on firm performance and conducting multiple regressions on family ownership involvement (FO), family management involvement (MO), and firm performance and establishing Models 1–2:

$$\text{Tobinq} = \alpha_0 + \alpha_1 \text{FO} + \sum \beta_i \text{Controls} + \varepsilon \tag{1}$$

$$\text{Tobinq} = \alpha_0 + \alpha_1 \text{MO} + \sum \beta_i \text{Controls} + \varepsilon \tag{2}$$

Step two: in order to test the moderating effect of family members' overseas experience and corporate charitable giving behavior on family involvement and corporate performance, models 3–6 were established:

$$\text{Tobinq} = \gamma_0 + \gamma_1 \text{FO} + \gamma_2 \text{Oversea} + \gamma_3 \text{FO} \times \text{Oversea} + \sum \beta_i \text{Controls} + \varepsilon \tag{3}$$

$$\text{Tobinq} = \gamma_0 + \gamma_1 \text{MO} + \gamma_2 \text{Oversea} + \gamma_3 \text{MO} \times \text{Oversea} + \sum \beta_i \text{Controls} + \varepsilon \tag{4}$$

$$\text{Tobinq} = \gamma_0 + \gamma_1 \text{FO} + \gamma_2 \text{Charity} + \gamma_3 \text{FO} \times \text{Charity} + \sum \beta_i \text{Controls} + \varepsilon \tag{5}$$

$$\text{Tobinq} = \gamma_0 + \gamma_1 \text{MO} + \gamma_2 \text{Charity} + \gamma_3 \text{MO} \times \text{Charity} + \sum \beta_i \text{Controls} + \varepsilon \tag{6}$$

## 4. Empirical Results and Analysis

### 4.1. Descriptive Statistics

It can be seen from the descriptive statistical results that the maximum value of the explained variable Tobin-q of the observed sample is 25.07, the minimum value is 0.717, and the mean value is 2.281, which means that there is an obvious gap between the performance of family enterprises. However, there is a certain difference in the degree of family ownership and management involvement of the sample enterprises. In terms of the degree of ownership, the maximum value is 0.899, the minimum value is 0.0525, and the mean value is 0.399, indicating that the shareholding ratio of family members in the sample enterprises varies from 5–90%. In terms of the involvement degree of family management rights, the maximum value is 0.900, the minimum value is 0.161, and the mean value is 0.436, indicating that the involvement of management rights in different family enterprises is significantly different and the separation system of two rights in family enterprises is obviously affected by the principal-agent problem. The average asset–liability ratio is 38.6%, indicating that the average debt level of the observation sample is moderate. The mean value of equity balance is 0.850, indicating that the family equity is highly concentrated. In addition, in order to avoid the multicollinearity problem, a variance inflation factor test was carried out for all variables in this paper. The results showed that the VIF values were all much less than 10, indicating that there was no collinearity problem among the variables (Table 2).

**Table 2.** Descriptive statistics of variables.

| Variable | Number of Observations | Mean | Standard Deviation | Minimum | Maximum | VIF |
|---|---|---|---|---|---|---|
| Tobin-q | 3696 | 2.281 | 1.622 | 0.717 | 25.07 | — |
| FO | 3696 | 0.399 | 0.167 | 0.0525 | 0.899 | 6.71 |
| MO | 3696 | 0.436 | 0.161 | 0.116 | 0.900 | 6.66 |
| Oversea | 3696 | 0.341 | 0.474 | 0 | 1 | 1.03 |
| Charity | 3696 | 0.490 | 0.524 | 0 | 1 | 1.14 |
| Size | 3696 | 22.00 | 1.087 | 19.11 | 27.01 | 1.44 |
| Lev | 3696 | 0.386 | 0.176 | 0.00836 | 0.990 | 1.36 |
| Age | 3696 | 3.022 | 0.239 | 2.197 | 3714 | 1.03 |
| Growth | 3696 | 0.293 | 1.342 | −11.68 | 33.42 | 1.04 |
| Index | 3696 | 9.699 | 1.737 | −1.420 | 12 | 1.02 |
| Ebd | 3696 | 0.850 | 0.616 | 0.0116 | 4 | 1.07 |
| Inde | 3696 | 0.00132 | 0.00832 | 0.00072 | 0.219 | 1.03 |

Table 3 reports the correlation coefficients of the main variables in this paper. As can be seen, the enterprise performance is positively correlated with the involvement of family members' ownership, as well as the involvement of family members' management, which is significant at the significance level of 1%. Several control variables selected in the study have significant relationships with independent variables and dependent variables and generally control other influencing factors that may affect dependent variables.

**Table 3.** The correlation coefficient of the variables.

| | Tobin-q | FO | MO | Oversea | Charity | Size | Lev | Age | Growth | Index | Ebd | Inde |
|---|---|---|---|---|---|---|---|---|---|---|---|---|
| Tobin-q | 1.000 | | | | | | | | | | | |
| FO | 0.150 *** | 1.000 | | | | | | | | | | |
| MO | 0.128 *** | 0.918 *** | 1.000 | | | | | | | | | |
| Oversea | 0.003 | 0.010 | 0.027 | 1.000 | | | | | | | | |
| Charity | 0.508 *** | 0.109 *** | 0.099 *** | 0.128 *** | 1.000 | | | | | | | |
| Size | −0.268 *** | −0.171 *** | −0.107 *** | 0.019 | −0.281 *** | 1.000 | | | | | | |
| Lev | −0.263 *** | −0.108 *** | −0.082 *** | 0.008 | −0.240 *** | 0.497 *** | 1.000 | | | | | |
| Age | −0.077 *** | −0.067 *** | −0.055 *** | 0.056 *** | −0.046 *** | 0.080 *** | 0.022 | 1.000 | | | | |
| Growth | −0.041 ** | −0.003 | −0.002 | −0.005 | −0.037 ** | 0.058 *** | 0.061 *** | 0.016 | 1.000 | | | |
| Index | 0.038 ** | 0.095 *** | 0.091 *** | 0.023 | 0.045 *** | −0.039 ** | 0.004 | −0.043 *** | −0.019 | 1.000 | | |
| Ebd | 0.038 ** | −0.142 *** | −0.206 *** | −0.030 * | 0.022 | −0.002 | −0.004 | −0.009 | 0.029 * | 0.036 *** | 1.000 | |
| Inde | −0.028 * | −0.048 *** | −0.047 *** | 0.074 *** | 0.036 ** | 0.080 *** | 0.086 *** | 0.107 *** | −0.008 | 0.017 | −0.029 * | 1.000 |

Notes: *, **, and ***, respectively, 10%, 5%, and 1% significance levels.

### 4.2. Multiple Regression Results

1. Research on the correlation between family involvement and enterprise performance

According to the regression analysis of Model 1 in Table 4, the correlation coefficient between family member ownership and enterprise performance is 1.035, which is significantly positive at the level of 1%, indicating that family member ownership will help promote corporate governance, improve market value, and improve company performance. The more ownership the family manager obtains, the better the performance of the family business will be in the intergenerational succession stage, meaning hypothesis H1a is supported.

According to the regression analysis of Model 2 in Table 4, the correlation coefficient between the management involvement of family members and enterprise performance is 1.048, which is significantly positive at the level of 1%, indicating that the management involvement will contribute to the implementation of better strategic decision-making in family enterprises. The more management rights obtained by family members, the better performance of the family business will be in the intergenerational transmission stage, meaning hypothesis H1b is supported.

In terms of control variables, the performance of the above two models is basically the same. The enterprise size, asset–liability ratio, enterprise age, and the degree of equity balance are significantly correlated with the performance of family enterprises at the level of 1%. This indicates that family listed companies with larger scales, longer establishment times, higher ownership concentrations, and good corporate credit have better performances.

2. Moderating effect of overseas experience of family members and charitable donation of family business

In order to verify the regulating effect of family members' overseas experiences, this paper introduces the crossing term between ownership and overseas experience (FO×Oversea) and the crossing term between management right and overseas experience (MO×Oversea), respectively, to obtain the corresponding Model 3 and Model 4. According to the regression analysis of column 3 in Table 4, the regression coefficient of the cross-multiplication term is −1.038 and it is significantly negatively correlated at the 1% level. According to the regression analysis of column 4 in Table 4, the regression coefficient of the cross-multiplication term is −0.971 and it is significantly negatively correlated at the 1% level. The results of the two models show that overseas study or work experience of family members is not conducive to improving the short-term performance of the original family company. Tobin-q is used as a measure of corporate performance and reflects the expected future profits of the market. Family executives with overseas backgrounds

will have strategic behaviors of cross-industry transformation or merger with the help of the overseas management experience learned, which is not conducive to the short-term development of enterprises. Therefore, H2a and H2b are assumed to be supported.

**Table 4.** Regression results.

| Variables | Model 1 | Model 2 | Model 3 | Model 4 | Model 5 | Model 6 |
|---|---|---|---|---|---|---|
| | Tobin-q | Tobin-q | Tobin-q | Tobin-q | Tobin-q | Tobin-q |
| FO | 1.035 *** | | 1.400 *** | | 0.097 | |
| | (6.64) | | (7.32) | | (0.51) | |
| MO | | 1.048 *** | | 1.380 *** | | 0.468 ** |
| | | (6.47) | | (6.98) | | (2.32) |
| Oversea | | | 0.448 *** | 0.454 *** | | |
| | | | (3.27) | (2.94) | | |
| Charity | | | | | 0.874 *** | 1.178 *** |
| | | | | | (7.66) | (8.99) |
| Size | −0.237 *** | −0.248 *** | −0.238 *** | −0.249 *** | −0.101 *** | −0.109 *** |
| | (−8.77) | (−9.22) | (−8.80) | (−9.25) | (−4.12) | (−4.48) |
| Lev | −1.581 *** | −1.575 *** | −1.585 *** | −1.576 *** | −0.960 *** | −0.996 *** |
| | (−9.53) | (−9.48) | (−9.56) | (−9.50) | (−6.44) | (−6.67) |
| Age | −0.358 *** | −0.364 *** | −0.380 *** | −0.380 *** | −0.268 *** | −0.275 *** |
| | (−3.37) | (−3.42) | (−3.57) | (−3.57) | (−2.84) | (−2.90) |
| Growth | −0.025 | −0.025 | −0.025 | −0.025 | −0.015 | −0.016 |
| | (−1.35) | (−1.34) | (−1.30) | (−1.33) | (−0.92) | (−0.94) |
| Index | 0.016 | 0.016 | 0.018 | 0.018 | 0.005 | 0.005 |
| | (1.11) | (1.12) | (1.21) | (1.20) | (0.36) | (0.41) |
| Ebd | 0.138 *** | 0.155 *** | 0.139 *** | 0.157 *** | 0.110 *** | 0.113 *** |
| | (3.32) | (3.68) | (3.35) | (3.74) | (2.98) | (3.03) |
| Inde | 2.178 | 2.298 | 1.770 | 1.785 | −3.870 | −3.826 |
| | (0.71) | (0.75) | (0.58) | (0.58) | (−1.42) | (−1.40) |
| FO × Oversea | | | −1.038 *** | | | |
| | | | (−3.29) | | | |
| MO × Oversea | | | | −0.971 *** | | |
| | | | | (−2.95) | | |
| FO × Charity | | | | | 1.346 *** | |
| | | | | | (5.01) | |
| MO × Charity | | | | | | 0.503 * |
| | | | | | | (1.79) |
| Constant | 8.515 *** | 8.711 *** | 8.424 *** | 8.605 *** | 4.807 *** | 4.873 *** |
| | (12.75) | (13.15) | (12.62) | (12.98) | (7.90) | (8.03) |
| Observations | 3696 | 3696 | 3696 | 3696 | 3696 | 3696 |
| R-squared | 0.111 | 0.110 | 0.114 | 0.112 | 0.299 | 0.294 |
| r2_a | 0.109 | 0.108 | 0.111 | 0.110 | 0.297 | 0.292 |
| F | 57.45 | 57.14 | 47.19 | 46.69 | 156.9 | 153.1 |

Notes: Numbers in parentheses are robust standard errors. *, **, and ***, respectively, 10%, 5%, and 1% significance levels.

In order to verify the regulating effect of family business charitable donations, this paper introduces the intersection term of ownership involvement and charitable donation (FO × Charity) and the intersection term of management involvement and charitable donation (MO × Charity), respectively, to obtain corresponding Model 5 and Model 6. According to the analysis of the regression results in column 5 of Table 4, the regression coefficient of the interaction term is 1.346 and has a significant positive correlation at the 1% level, indicating that family enterprises with charitable donation behaviors are more socially responsible and ownership involvement can promote enterprises to improve performance and perform social responsibility more actively. Therefore, hypothesis H3a is supported. According to the regression analysis of column 6 in Table 4, the regression coefficient of the interaction term is 0.503 and the correlation is significantly positive at the 10% level, indicating that philanthropic activities can help them better manage the enterprise and build the brand effect. The involvement of management rights can improve the performance of the enterprise. Consequently, hypothesis H3b is supported.

### 4.3. Test for Robustness

In order to ensure the robustness of the empirical results in this paper, the following methods are used to conduct robustness tests:

1.  Substitution variable method

In this paper, ROA is used to replace the Tobin-q to measure corporate performance. The regression results are shown in Table 5. As can be seen, in Model 1, family ownership involvement is significantly positively correlated with firm performance at the level of 1%, which again supports hypothesis H1a. In Model 2, the management involvement of family firms was positively correlated with firm performance at the 1% level, which supported hypothesis H1b. The coefficient of the Model 3 intersection term is significantly negative at the level of 1%, which supports hypothesis H2a. The coefficient of the Model 4 intersection term is significantly negative at the level of 1%, which supports hypothesis H2b. The coefficient of Model 5 intersection term is significantly positive at 5% level, which supports hypothesis H3a. The coefficient of Model 6 intersection term is significantly positive at the 10% level, which supports hypothesis H3b.

**Table 5.** Robustness check—substitution variables.

| Variables | Model 1 | Model 2 | Model 3 | Model 4 | Model 5 | Model 6 |
|---|---|---|---|---|---|---|
| | ROA | ROA | ROA | ROA | ROA | ROA |
| FO | 0.186 *** | | 0.218 *** | | 0.141 *** | |
| | (12.06) | | (11.67) | | (6.72) | |
| MO | | 0.195 *** | | 0.229 *** | | 0.157 *** |
| | | (12.13) | | (11.86) | | (7.13) |
| Oversea | | | 0.038 *** | 0.045 *** | | |
| | | | (2.94) | (3.05) | | |
| Charity | | | | | 0.022 * | 0.027 * |
| | | | | | (1.80) | (1.88) |
| Size | 0.039 *** | 0.038 *** | 0.039 *** | 0.037 *** | 0.044 *** | 0.042 *** |
| | (14.71) | (14.09) | (14.36) | (13.74) | (16.13) | (15.48) |
| Lev | −0.248 *** | −0.247 *** | −0.248 *** | −0.247 *** | −0.225 *** | −0.225 *** |
| | (−15.10) | (−15.02) | (−15.05) | (−14.96) | (−13.66) | (−13.69) |
| Age | −0.010 | −0.011 | −0.013 | −0.013 | −0.008 | −0.009 |
| | (−0.96) | (−1.04) | (−1.21) | (−1.26) | (−0.74) | (−0.83) |
| Growth | 0.005 *** | 0.005 *** | 0.005 *** | 0.005 *** | 0.005 *** | 0.005 *** |
| | (2.58) | (2.59) | (2.63) | (2.61) | (2.84) | (2.84) |
| Index | 0.001 | 0.001 | 0.001 | 0.001 | 0.000 | 0.000 |
| | (0.53) | (0.51) | (0.59) | (0.57) | (0.22) | (0.24) |
| Ebd | −0.000 | 0.003 | −0.000 | 0.003 | −0.001 | 0.002 |
| | (−0.04) | (0.77) | (−0.02) | (0.82) | (−0.25) | (0.46) |
| Inde | 0.074 | 0.101 | 0.751 ** | 0.752 ** | 0.646 ** | 0.696 ** |
| | (0.25) | (0.33) | (2.42) | (2.42) | (2.14) | (2.30) |
| FO × Oversea | | | −0.089 *** | | | |
| | | | (−2.98) | | | |
| MO × Oversea | | | | −0.097 *** | | |
| | | | | (−3.13) | | |
| FO × Charity | | | | | 0.073 ** | |
| | | | | | (2.50) | |
| MO × Charity | | | | | | 0.054 * |
| | | | | | | (1.78) |
| Constant | −0.752 *** | −0.721 *** | −0.741 *** | −0.714 *** | −0.862 *** | −0.831 *** |
| | (−11.37) | (−11.00) | (−11.15) | (−10.82) | (−12.85) | (−12.45) |
| Observations | 3696 | 3696 | 3696 | 3696 | 3696 | 3696 |
| R-squared | 0.106 | 0.107 | 0.108 | 0.108 | 0.131 | 0.131 |
| r2_a | 0.104 | 0.105 | 0.105 | 0.106 | 0.129 | 0.128 |
| F | 54.77 | 55.01 | 44.50 | 44.78 | 55.78 | 55.35 |

Notes: Numbers in parentheses are robust standard errors. *, **, and ***, respectively, 10%, 5%, and 1% significance level.

## 5. Conclusions and Implications

### 5.1. Main Conclusions

Using A-share family listed companies in Shanghai and Shenzhen from 2018 to 2020 as research samples, this study mainly investigates the influence effect of family involvement on corporate performance, as well as the moderating effect between members' overseas experience and corporate charitable giving behavior. The main conclusions of this paper are as follows: (1) family ownership involvement can significantly improve the performance of family firms; (2) the involvement of the second-generation family management right can significantly improve enterprise performance; (3) the overseas experience of second-generation members can negatively regulate the relationship between family involvement and enterprise performance; and (4) the charitable donation behavior of family firms can positively regulate the relationship between family involvement and firm performance.

### 5.2. Research Implications

First of all, for a family business, in order to achieve the goal of starting a business for one generation and keeping the business for the second generation, it is necessary to formulate a business strategy suitable for the sustainable development of the company: strengthen property rights governance, clarify property rights, and allocate property rights scientifically and rationally, so that corporate interests and the interests of the employees are closely linked; perfect the internal management mechanism of the enterprise and use an effective internal formal management system to help employees improve work efficiency and concentrate on laying a good foundation for the sustainable development of the enterprise; build a good corporate culture, combine the actual situation, and form an effective gathering of members from outside the family within the company to create a harmonious whole; and formulate the successor plan of the family business in advance to avoid unnecessary family turmoil caused by emergencies. For family businesses, upgrading, transformation, intergenerational inheritance, and common prosperity are the main problems at this stage. Therefore, family businesses must actively respond to various challenges brought about by globalization in the course of operation and, during the transition period, family members should unite and work hard to maximize corporate wealth and improve corporate performance.

Secondly, for government departments, it is imperative to improve the corporate information disclosure system and preferential policies. Formulate and issue relevant documents, try to implement a punishment and reward mechanism for the performance of small- and medium-sized enterprises, and promote the vigorous development of private enterprises represented by family enterprises: such as implementing appropriate tax reductions and exemption measures for family enterprises with high corporate performances. In order to further promote the sustainable development of private enterprises, the government should publicly commend entrepreneurs who actively implement charitable donations. In this way, family members can increase their recognition of entrepreneurs and enterprises and realize the inheritance of family social and emotional wealth. When recommending deputies to the National People's Congress, members of the CPPCC, party representatives, or other candidates who can enhance the political status of entrepreneurs, local governments can stipulate that corporate performance and total charitable donations reach a certain threshold, so as to encourage private entrepreneurs and families to commit to sustainable development. At the same time, it provides real and reliable data support for various stakeholders, forming a complete logic chain of "policy orientation-capital market investment-high-quality development of family business".

### 5.3. Research Limitations and Future Research Directions

The limitations of this paper are as follows: (1) Using family listed companies as the research object, the situations of non-listed family companies are not taken into account. In reality, many unlisted family enterprises in China also have the problem of power allocation of family members. However, due to the difficulty of obtaining data, this paper

fails to discuss the situation of unlisted family companies. Future research can enrich the research of unlisted family enterprises through field research and visits. (2) The quantitative research on the influence of family involvement on firm performance was carried out using secondary data, but the research on its mechanism was still not in-depth. Future research can combine the scales in the field of psychology and organizational behavior to conduct a questionnaire survey on family enterprises with overseas experience and charitable donation behavior. All the above need to be deepened and expanded in future research.

**Author Contributions:** M.W. and S.S. contributed to the conception of the study; M.W. and Y.Z performed the experiment; Y.Z. and Y.L. contributed significantly to analysis and manuscript preparation; M.W. and S.S. performed the data analyses and wrote the manuscript; Y.Z. and Y.L. helped perform the analysis with constructive discussions. All authors have read and agreed to the published version of the manuscript.

**Funding:** This work was funded by the National Social Science Foundation of China (19BGL127), the National Natural Science Foundation of China (72072076), and the Fund for Humanities and Social Sciences Research of the Ministry of Education (18YJA630074).

**Institutional Review Board Statement:** Not applicable.

**Informed Consent Statement:** Not applicable.

**Data Availability Statement:** The data presented in this article are available from the corresponding author upon request.

**Conflicts of Interest:** All authors declare that they have no conflict of interest.

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
