# Peer review of "Can Family Involvement Improve Business Performance? Based on the Dual Moderating Effects of Overseas Experience and Charitable Donations"

_sustainability, doi:10.3390/su142316141_

Round 1
Reviewer 1 Report
The paper focuses on a really interesting and relevant, I hope the following recommendation could help improve its quality.
I would suggest to expand the bibliographical section to give a better e more complete framework of this specific kind of organizations and their dynamics
I would suggest (if possible) to give further details about the organizations involved (i.e. is it possible to group them by specific characteristics such as business type? Moreover it would really interesting verify if those characteristics lead to differences in results)
I would suggest to expand the discussion of results by enriching the argument with more connections with respect to theory and implications for practionners
Author Response
Comment 1: I would suggest to expand the bibliographical section to give a better e more complete framework of this specific kind of organizations and their dynamics
Response: Thank you for your suggestion, I added more references to the article
Chen Ling and Ying Lifen (2003) Hereditary Succession: The Inheritable Management and Creation in Clannish Enter-prises [J]. Management World (06):89-97+155-156.
Jiang Tao;Yang Mingxuan;Wang Han (2019) Institutional Environment, Second-generation Involvement and Objective Duality: An Empirical Study from Chinese Listed Family Firms[J]. Nankai Business Review.22(04):135-147.
Li Xinchun;Ma Jun;He Xuan;Yuan Yuan(2018) The Modern Transformation of Family Governance: Coevolve of Family Involvement and Family Formal Institution[J]. Nankai Business Review.21(02):161-170.
Luo Siping and Yu Yongda(2012)Technology Transfer, "Returnees" and Enterprise Technology Innovation——An Empirical Study Based on China's Photovoltaic Industry[J]. Management World 11:124-132.
Shao Yinghong;Ding Qin;Bao Qing(2021) CEO Power Intensity and Enterprises' Ambidexterity Innovation Investment: the Moderating Role of Marketization Level and Internal Control[J]. Science & Technology Progress and Policy.39:1-9.
Suo Jian;Yang Han(2021) The Response of Capital Market to Involvement of Enterprise Successors from the Perspective of Intrapreneurship[J]. Contemporary Economic Management. 12:1-24.
Yang Xuelei; Li Weining; Shang Hangbiao(2022) Study on the Influence of Characteristics of Overseas Experience of Second Generations on Portfolio Entrepreneurship in Family Enterprises: from the Perspective of Intergenerational Conflict[J]. Science & Technology Progress and Policy.(39):1149-1169.
Lumpkin G T, Brigham K H.(2011). Long-Term Orientation and Intertemporal Choice in Family Firms[J]. Entrepre-neurship Theory and Practice, 35:1149-1169.
Oradi J, Asiaei K, Rezaee Z.(2020). CEO financial background and internal control weaknesses[J]. Corporate Govern-ance: An International Review, 28:119-140.
YANG Zhen; MA Guangyuan; CHEN Jin(2021) Entrepreneurs Comprehensive Status, Family Involvement and Corpo-rate Social Responsibility——Micro Evidence from the Survey of Private Enterprises in China[J]. Economic Perspec-tives,8:101-115.
ZOU Likai; LIANG Qiang; WANG Bo(2019) An Empirical Study on Second-Generation Succession Mode Based on the Perspective of Authority Transformation[J]. Chinese Journal of Management,16(12):1771-1780+1789.
Xu Mengna; Zhou Shengchun(2008)Research on Agency Problems and Intergenerational Inheritance of Family Enterprises Based on the Perspective of Altruism[J]. Foreign Economics & Management,7:52-57
WU Meng-yun; ZHANG Lin-rong(2018) Research on Attributes of Top Management Team, Environmental Responsibility and Corporate Value[J]. East China Economic Management,32:122-129
Miller, Isabelle and Richard(2010). Family ownership and acquisition behavior in publicly-traded companies [J]. Stra-tegic Management Journal, 31(2): 201-223.
Liang Qiang; Zhang Jiamei; Lin Jindan(2022). How dose Family Management Involvement Affect M&A Strategic Be-havior: Based on the Empirical Research of Chinese Listed Family Firms[J]. South China Journal of Economics. (06):118-136
Su Qilin,Zhu Wen(2003). Family Control and Firm Value: Evidence from China Listed Companies[J]. Economic Re-search Journal, (08):36-45+91.
Comment 2: I would suggest (if possible) to give further details about the organizations involved (i.e. is it possible to group them by specific characteristics such as business type? Moreover it would really interesting verify if those characteristics lead to differences in results)
Response: Thank you for your suggestion. Our research object is the situation of family business listed companies, which is suitable for full sample analysis. If the sample size is subdivided into industries, it is difficult to support our mainstream views. In future research, we will combine your suggestions and consider using industry classification to explore Sustainable development of family business, thank you for providing us with a new idea
Comment 3: I would suggest to expand the discussion of results by enriching the argument with more connections with respect to theory and implications for practitioners
Response: Thank you for your advice. We think to expand some theory in the ‘Literature and research hypotheses’. While family involvement injects special vitality into the enterprise, it is also accompanied by conflicts of interest. With the deepening of the involvement of family management power, special agency problems mixed with internal goal conflicts of family enterprises have gradually emerged. The principal-agent theory holds that due to information asymmetry between the owner and the operator, there are conflicts between the two parties in terms of goals and motivations (Su et al. 2003). At the same time, the high concentration of family business ownership and management rights has weakened the traditional agency problem (Miller et.al., 2010). Based on the principal-agent theory, self-agency in the context of family business still has conflicts, and decision preferences derived from multiple objectives will lead to Differences at the level of corporate strategy (Liang et.al., 2022)
About “implications for practionners”, we expand the implication for family business. For a family business, in order to achieve the goal of starting a business for one generation and keeping the business for the second generation, it is necessary to formulate a business strategy suitable for the sustainable development of the company: strengthen property rights governance, clarify property rights and allocate property rights scientifically and rationally, so that corporate interests and The interests of employees are closely linked; perfect the internal management mechanism of the enterprise, and an effective internal formal management system can help employees improve work efficiency and concentrate on laying a good foundation for the sustainable development of the enterprise; build a good corporate culture, combine the actual situation and form an effective The unique corporate culture gathers members outside the family within the company to create a harmonious whole together; formulate the successor plan of the family business in advance to avoid unnecessary family turmoil caused by emergencies. For family businesses, upgrading and transformation, intergenerational inheritance and common prosperity are the main problems at this stage. Therefore, family businesses must actively respond to various challenges brought about by globalization in the course of operation, and during the transition period, family members should unite and work hard to maximize corporate wealth and improve corporate performance.
Reviewer 2 Report
Dear Authors:
Sustainable development of private enterprises is an eternal topic. This study finds that family involvement can significantly improve enterprise performance, and overseas experience and charitable donation behavior have a moderating effect on it. However, there are some questions I would like to discuss with the author(s):
1.In the introduction, the author mentions "providing relevant countermeasures for the future sustainable development of family enterprises for reference", but it lacks some discussion on the importance of sustainable development to family enterprises, and the research contribution in the introduction is not clear enough. It is suggested to revise and refine the research contribution.
2. The research revelation is not clear, so it is suggested to expand and subdivide the dimensions. Please reconsider revising the writing with better organization.
3. Are the criteria for defining family involvement diverse? Why did the author choose the ownership and management dimensions? Are there more documentary theories to support it?
Author Response
Comment 1: In the introduction, the author mentions "providing relevant countermeasures for the future sustainable development of family enterprises for reference", but it lacks some discussion on the importance of sustainable development to family enterprises, and the research contribution in the introduction is not clear enough. It is suggested to revise and refine the research contribution.
Response: In response to your suggestion, I cite typical family cases to tell the importance of sustainable development of family businesses, with some modifications for research contributions.
As one of the world's top 20 pharmaceutical companies, Boehringer Ingelheim, a family-owned pharmaceutical company, relies on its outstanding performance in the fields of long-term strategic positioning, organizational culture, and talent development to achieve the goal of "becoming rich for three generations". Employees' long-term practice of family values of sustainable development and the driving effect of a diverse and inclusive organizational culture on innovation. The vision of long-term sustainable development is an important force to enhance organizational resilience in the face of uncertain environments.
Research contribution: Secondly, through empirical research on the mechanism of the second generation's involvement in ownership and management rights affecting the performance of family businesses, it is found that the degree of contribution of the second generation's involvement in family businesses and charitable donations can positively promote the development of family businesses, and effective charitable Donations can better help family businesses establish their image, win reputation, and gain social recognition.
Comment 2: The research revelation is not clear, so it is suggested to expand and subdivide the dimensions. Please reconsider revising the writing with better organization.
Response: Thank you for your suggestion. I subdivide the research revelation into three aspects:
First of all, for a family business, in order to achieve the goal of starting a business for one generation and keeping the business for the second generation, it is necessary to formulate a business strategy suitable for the sustainable development of the company: strengthen property rights governance, clarify property rights and allocate property rights scientifically and rationally, so that corporate interests and The interests of employees are closely linked; perfect the internal management mechanism of the enterprise, and an effective internal formal management system can help employees improve work efficiency and concentrate on laying a good foundation for the sustainable development of the enterprise; build a good corporate culture, combine the actual situation and form an effective The unique corporate culture gathers members outside the family within the company to create a harmonious whole together; formulate the successor plan of the family business in advance to avoid unnecessary family turmoil caused by emergencies. For family businesses, upgrading and transformation, intergenerational inheritance and common prosperity are the main problems at this stage. Therefore, family businesses must actively respond to various challenges brought about by globalization in the course of operation, and during the transition period, family members should unite and work hard to maximize corporate wealth and improve corporate performance.
Secondly, for government departments, it is imperative to improve the corporate information disclosure system and preferential policies. Formulate and issue relevant documents, try to implement a punishment and reward mechanism for the performance of small and medium-sized enterprises, and promote the vigorous development of private enterprises represented by family enterprises: such as implementing appropriate tax reduction and exemption measures for family enterprises with high corporate performance. In order to further promote the sustainable development of private enterprises, the government publicly commends entrepreneurs who actively implement charitable donations. In this way, family members can increase their recognition of entrepreneurs and enterprises, and realize the inheritance of family social and emotional wealth. When recommending deputies to the National People's Congress, members of the CPPCC, party representatives, or other candidates who can enhance the political status of entrepreneurs, local governments can stipulate that corporate performance and total charitable donations reach a certain threshold, so as to encourage private entrepreneurs and families to commit to sustainable development. At the same time, it provides real and reliable data support for various stakeholders, forming a complete logic chain of "policy orientation-capital market investment-high-quality development of family business".
Comment 3: Are the criteria for defining family involvement diverse? Why did the author choose the ownership and management dimensions? Are there more documentary theories to support it?
Response: Thank you for your suggestion. Regarding the independent variable standards mentioned in this article, we have made the following supplementary explanations:
Family involvement, as one of the distinctive features of family business, refers to the continuous investment of human capital, cultural capital, social capital and financial capital in the family business by the family, and different control over the ownership, control, operation, income, culture, etc. of the family business. and influence, and produce a series of behaviors or concepts involving family members or family interest communities in the family business, which ultimately affects the growth and development of the business. There are two ways of family involvement, formal involvement and informal involvement. Formal involvement refers to ownership involvement, control right involvement and operation (management) involvement, etc. Informal involvement includes family willingness involvement, family spirit involvement, etc. Involvement and family culture involvement. Scholars at home and abroad generally use formal family involvement as indicators to measure. Chen Ling and others believe that family involvement mainly refers to the involvement of ownership and management rights, which generally increases the complexity and uncertainty of family business development, and may cause Lead to fundamental changes in corporate structure, culture and systems. On the other hand, Jiang Tao et al. (2019) judged that when a family member or several family members’ children, son-in-law, nephew (daughter), and nephew (daughter) appeared in the enterprise, family involvement behavior occurred, marked as 1, Otherwise 0. Combining two common research methods, the researchers believe that the situation of family involvement includes multiple combinations such as husband and wife joint management, father-son (female) inheritance, etc., combined with the situation of family business ownership and control, the shares of all family members in the company The ratio of family ownership obtained from the sum of the sum/total shares of the enterprise and the ratio of family management rights obtained from the number of family members in management/total number of managers are more in line with the actual situation.
Round 2
Reviewer 1 Report
I appreciate the improvement of the quality of the paper